# An Evaluation of Community Health Workers’ Knowledge, Attitude and Personal Lifestyle Behaviour in Non-Communicable Disease Health Promotion and Their Association with Self-Efficacy and NCD-Risk Perception

**DOI:** 10.3390/ijerph20095642

**Published:** 2023-04-25

**Authors:** Melaku Kindie Yenit, Tracy L. Kolbe-Alexander, Kassahun Alemu Gelaye, Lemma Derseh Gezie, Getayeneh Antehunegn Tesema, Solomon Mekonnen Abebe, Telake Azale, Kegnie Shitu, Prajwal Gyawali

**Affiliations:** 1School of Health and Medical Sciences, Centre for Health Research, University of Southern Queensland, Ipswich, QLD 4305, Australia; 2Department of Epidemiology and Biostatistics, Institute of Public Health, College of Medicine and Health Sciences, University of Gondar, Gondar P.O. Box 196, Ethiopia; 3Division of Exercise Science and Sports Medicine, Department of Human Biology, Faculty of Health Sciences, University of Cape Town, Cape Town 7700, South Africa; 4Department of Human Nutrition, Institute of Public Health, College of Medicine and Health Sciences, University of Gondar, Gondar P.O. Box 196, Ethiopia; 5Department of Health Promotion and Health Behavior, Institute of Public Health, College of Medicine and Health Sciences, University of Gondar, Gondar P.O. Box 196, Ethiopia; 6Centre for Health Research, School of Health and Medical Sciences, University of Southern Queensland, Toowoomba, QLD 4350, Australia

**Keywords:** non-communicable disease, knowledge, attitude, lifestyle, community health workers, Ethiopia

## Abstract

Community health workers, also known as health extension workers (HEWs), play an important role in health promotion. This study evaluates HEWs’ knowledge, attitude, and self-efficacy for non-communicable diseases (NCD) health promotion. HEWs (n = 203) completed a structured questionnaire on knowledge, attitude, behaviour, self-efficacy and NCD risk perception. Regression analysis was used to determine the association between self-efficacy and NCD risk perception with knowledge (high, medium, low), attitude (favourable/unfavourable) and physical activity (sufficient/insufficient). HEWs with higher self-efficacy were more likely to have high NCD knowledge (AOR: 2.21; 95% CI: 1.21. 4.07), favourable attitude towards NCD health promotion (AOR: 6.27; 95% CI: 3.11. 12.61) and were more physically active (AOR: 2.27; 95% CI: 1.08. 4.74) than those with lower self-efficacy. HEWs with higher NCD susceptibility (AOR: 1.89; 95% CI: 1.04. 3.47) and perceived severity (AOR: 2.69; 95% CI: 1.46, 4.93) had higher odds of NCD knowledge than their counterparts. Moreover, sufficient physical activity was influenced by HEWs’ perceived NCD susceptibility and perceived benefits of lifestyle change. Therefore, HEWs need to adopt healthy lifestyle choices to become effective role models for the community. Our findings highlight the need to include a healthy lifestyle when training HEWs, which might increase self-efficacy for NCD health promotion.

## 1. Introduction

Non-communicable diseases (NCDs) cause approximately 41 million deaths annually, accounting for 74% of all deaths worldwide [1]. Ethiopia is a low-income country in the Global South, experiencing demographic, nutritional, and epidemiological transitions favouring the rise in NCDs [2,3]. One-third of Ethiopians have been diagnosed with an NCD, with cardiovascular diseases being the most frequent diagnosis [4].

Lifestyle behaviours, such as insufficient physical activity, unhealthy diets, tobacco use and harmful use of alcohol, are the major risk factors for NCDs [1,5]. These risk factors have contributed to the increased prevalence of NCDs, leading to a greater demand on an already overwhelmed healthcare system [3,5]. Hence, effective community-based NCD prevention and management strategies that focus on modifiable risk factors and promote a healthy lifestyle are required [1,6].

Most NCD health services in low–middle-income countries currently rely on primary healthcare services, with less emphasis on early prevention and management. Recent literature suggests that including community health workers in NCD-related health services is effective in prevention and control [7,8]. Community health workers are valuable resources in the global health workforce. They can lead and facilitate health education and counselling, promote healthy lifestyles, patient care, and social support, and facilitate liaising with the healthcare system [8,9]. As a result, several countries have started recognizing the need and incorporating community health workers into their healthcare systems [10].

The effectiveness of health workers in NCD health promotion is influenced mainly by their knowledge of NCDs and related risk factors, risk perception, and their self-efficacy to include health promotion as part of their tasks [11]. Many health behaviour models suggest that people’s perception of risk is a key factor in their willingness to change their behaviour [12]. Perception of risk and self-efficacy are predictors of healthy behaviour and people’s likelihood of engaging in preventive behaviours [13,14]. Therefore, understanding community health workers’ knowledge, attitude and behaviour and their association with NCD risk perception and self-efficacy can inform interventions to strengthen their effective participation in NCD health promotion.

Furthermore, community healthcare workers are seen as role models by the community and are expected to lead by example. Hence, these community healthcare workers engage in healthy behaviours, such as physical activity and a healthy diet, and improve their lifestyle [15,16,17].

The Health Extension Program is a flagship initiative of the Ethiopian government that was started by the Ministry of Health in 2003 to develop community health literacy focusing on infectious diseases and child and maternal health. This program uses a community healthcare worker model by employing Health Extension Workers (HEW). The HEWs’ role is to improve access to primary healthcare and alleviate the healthcare sector’s limited human resources. Despite the program’s success in reducing infectious diseases and improving maternal and child health, NCD-related health services in cities are not addressed adequately [18,19]. Consequently, HEW’s services are not being maximized [20].

Although HEWs are the pillars of Ethiopia’s healthcare system in terms of NCD health promotion, their knowledge, attitude, and behaviour towards NCD health promotion, their NCD risk perception and self-efficacy are unknown. Therefore, this study aimed to investigate HEWs’ level of knowledge, attitude, and personal lifestyle behaviours and their association with NCD-risk perception and self-efficacy in the north Gondar zone, northwest Ethiopia.

## 2. Materials and Methods

### 2.1. Setting

This study was conducted in three districts of the North Gondar zone in Ethiopia: Gondar, Debark and Dabat. With a population of almost three hundred thousand people, Gondar is one of Ethiopia’s largest cities and 780 km north of Ethiopia’s capital, Addis Ababa. Dabat and Debark are located 70 and 100 km from Gondar, respectively.

The HEW program was expanded in 2009 to address NCDs in urban areas. The program has both rural and urban HEWs. They are responsible for promoting health and providing primary healthcare services in their communities. Rural HEWs are recruited based on criteria such as residence in the village, ability to speak the local language, completion of the 10th grade, and a commitment to serve in the village [20]. Rural HEWs attend a year-long training program, including both theoretical and practical sessions. After graduation, they are employed in paid positions in health services. The urban HEWs are clinical nurses with diplomas who have undergone a three-month pre-service Health Extension Program training. More than 800 HEWs are currently employed in the North Gondar zone, of which 225 HEWs are employed in the Gondar, Debark and Dabat districts.

### 2.2. Study Design and Period

We conducted a cross-sectional study among HEWs working in three districts of the North Gondar zone from February to April 2022. The study was part of a larger project to improve HEWs’ capacity in NCD prevention and promote healthy lifestyle behaviour in the selected districts.

### 2.3. Sample Size and Participants

The required sample size was calculated using the single population proportion formulas of $n=\frac{{\left(z_{1-a/2}\right)}^{2}p\;\left(1-p\right)}{d^{2}}\;$ as used by Arya, Antonisamy et al. [21] with the following assumptions: fifty percent of HEW had some knowledge of NCD, a good attitude towards promoting a healthy lifestyle and optimum lifestyle practice, a 95% level of confidence, and a 5% margin of error (d), resulting in an ideal sample size of 385. A 50% proportion was considered because no similar previous studies have been conducted among HEWs in Ethiopia. However, the number of HEWs in the selected study area was less than the calculated sample size. Therefore, all available HEWs (*n* = 225) working in the selected districts were invited to participate in the study.

### 2.4. Data Collection Tool and Procedure

Data were collected using a structured, pretested, and self-administered questionnaire. The data collection tool was developed based on the WHO step-wise survey and after reviewing previous studies used by Samuel [22], Demaio et al. [23], Ji Catherine [24], and Mahajan et al. [25]. The questionnaire comprised of 6 sub-sections: (1) socio-demographic and personal lifestyle behaviour (physical activity and dietary habit); (2) general knowledge of NCDs; (3) knowledge of specific NCDs (hypertension, diabetes, and cardiovascular diseases); (4) attitude towards promoting healthy lifestyles; (5) NCD-risk perception; and (6) self-efficacy of promoting healthy lifestyles. These questionnaires are provided as the Appendix A. The questionnaire was prepared in English, then translated into the local language (Amharic), and back-translated to English by a language expert and public health professional. The questionnaire was pretested among 25 HEWs (10% of the sample size) outside the study area to ensure consistency. This study’s knowledge and attitude measurement tools were internally consistent, with Cronbach-alpha values of 0.71 and 0.744, respectively. Nine data collectors and four supervisors were involved in the data collection process. Data collectors received five days of training on the study’s objective, the confidentiality of information, and data collection methods. We contacted HEW via the respective health department, visited the health facility during working days, explained the study’s purpose and confidentiality of information, and asked for their consent to participate. We invited all HEWs in the study’s districts. Those HEWs interested in participating were included, but those unavailable during the data collecting time were excluded. The data collection process was conducted from February to April 2022.

### 2.5. Measures

#### 2.5.1. Socio-Demographic and Lifestyle Behaviours of HEWs

HEWs reported their place of residence, age, educational status, marital status, years of employment, and family history of NCDs.

##### Physical Activity

The International Physical Activity Questionnaire (IPAQ-7)-Short Form was used to quantify physical activity [26]. HEWs reported on the frequency and duration of moderate- and vigorous-intensity physical activity in the past week. They also reported on time spent sitting on a week and weekend day. The Metabolic Equivalent of Task (MET) minutes per week were estimated by multiplying the duration in minutes, frequency in days, and MET intensity [27]. HEWs who participated in 150 min or more of physical activity per week were categorized as sufficiently active [28].

##### Dietary Habits

HEWs reported on the frequency of fruit and vegetable consumption per week. Dietary habits were classified into two groups for this study based on participants’ consumption of fruits and vegetables. Those who consumed daily fruits and/or vegetable received a “1” score and were labelled as having adequate fruit and/or vegetable consumption, whereas those with less than a daily fruit and/or vegetable intake received a “0” score for having inadequate vegetable and/or fruit consumption, which is consistent with previous studies in Ethiopia [29,30].

#### 2.5.2. Knowledge, Attitudes and Perception of NCD

##### Knowledge of NCDs 

This section of the questionnaire was adapted from surveys previously used in Uganda [9], Mongolia [23], India [25], and Brazil [24]. We used a total of 75 questions to measure the level of comprehensive knowledge, and scores ranged from 0–75. Participants were asked to identify NCDs from a list of diseases, NCD risk factors and dietary recommendations for NCD patients. The options for responses were “Yes, no, or unsure/I do not know”.

Knowledge of diabetes, hypertension, and cardiovascular disease was assessed by asking HEWs about their general awareness, risk factors, diagnostic criteria, signs and symptoms, complications, preventive strategies, and management. The responses were “True or false, yes or no, and multiple choice”. Each correct answer was given a score of 1, and the incorrect answer was given a 0 score. The “I do not know” response was also scored zero as it reflects insufficient knowledge [31].

The main outcome measure was a comprehensive knowledge score, which was the average of all responses on general NCD knowledge and knowledge of specific NCDs. Participants’ knowledge was classified as low (tertile 1; <55/75), moderate (tertile 2; 55–59/75), or high (tertile 3; >60/75).

##### Attitude

HEWs’ attitudes towards advocating healthy lifestyles to the community were assessed using ten items, where their responses were recorded on a five-point Likert scale (strongly disagree—1 to strongly agree—5). The beliefs of the HEWs about their role in health promotion, the benefits of lifestyle promotion, their attitude towards people’s willingness to accept lifestyle promotion, and the barriers to promoting healthy lifestyles, were evaluated. A composite score was calculated by summing responses to the 10 items, and the aggregate score ranged from 10 to 50. Similar to Bitew, Sharew et al. [32], the median score was used to categorise attitudes as either ‘favorable’ or ‘unfavorable’ for promoting healthy lifestyles.

##### NCD Risk Perception

NCD risk perception questions were based on the Health Belief Model constructs, whereby perceived susceptibility, severity and benefits were assessed [12]. Each construct of the Health Belief Model was assessed by a series of questions, requiring responses on a five-point Likert scale (strongly disagree—1 to strongly agree—5). Perceived susceptibility assessed HEWs’ self-perception of their vulnerability to NCDs, measured by a summed score of 5 items, and possible scores ranged from 5 to 25. The HEWs’ perceived NCD severity was assessed using a summed score of 3 items, with possible scores ranging from 3 to 15. A perceived benefit of lifestyle was a HEW’s belief about the effectiveness of lifestyle measures as a strategy of NCD prevention, measured by a summed score of 6 items, possible scores ranging from 6 to 30. A single composite score was computed for each construct (perceived susceptibility, severity and benefits) based on the formula used by Tsegaw and colleagues [33] (highest score–lowest score)/2 + lowest score. A cut-off score of 15 ((22 − 8)/2 + 8 = 15) was used to classify participants into 2 groups as low and high perceived susceptibility. For perceived severity, we used a cut-off score of 9 ((15 − 3)/2 + 3 = 9), and for perceived benefits, a cut-off score of 18 ((30 − 6)/2 + 6 = 18) was used.

##### Perceived Self-Efficacy

Perceived self-efficacy assessed HEWs’ self-reported confidence in promoting healthy lifestyles, screening, and referring high-risk individuals, measuring blood pressure, and performing a rapid blood glucose test. HEWs were asked about their confidence in advising their clients about the benefits of regular exercise, healthy diets, the harms of cigarette and alcohol use, and their confidence in measuring blood glucose levels and blood pressure and supporting chronic patients. A single composite score was computed by summing the responses to the ten questions with a five-point Likert scale. A score possibly ranged from 10 to 50. This score was normally distributed, and therefore the mean was calculated and used as a cut-off to categorize participants into two groups: higher self-confidence and lower self-confident. This approach of classifying participants’ perceived self-efficacy into two groups is similar to previous studies [34].

### 2.6. Data Analysis

The data were coded and entered into IBM Statistical Package for the Social Sciences (SPSS) version 26 for analysis. Descriptive statistics, such as frequencies, percentages, mean (SD) or median (IQR), were computed. The Kolmogorov–Smirnov test was used to determine the normality of the data. A *p*-value of less than 0.05 at the Kolmogorov–Smirnov test was used to indicate that the data were not normally distributed. We used tertiles to categorize NCD knowledge as low, medium, and high. The median score was used to classify attitude as either favourable or unfavourable. We used the mean to classify perceived self-efficacy as higher or lower.

An ordinal logistic regression analysis was used to associate NCD risk perception and perceived self-efficacy with knowledge. The Brant test statistic was used to examine the proportional odds parallel line assumption of ordinal logistic regression analysis. A non-significant test statistic was used to prove that the parallel regression assumption was not violated. This indicates that the effect of any independent variables was constant across the different levels of comprehensive NCD knowledge categories (low, medium, and high). As a result, a single proportional odds ratio was used for interpretation. A binary logistic regression analysis model was used to associate NCD risk perception and perceived self-efficacy with attitude towards lifestyle promotion and lifestyle behaviour (physical activity). All these regression models were adjusted for the effect of age, level of education, and service years.

## 3. Results

### 3.1. Socio-Demographic and Lifestyle Behaviours of HEWs

#### 3.1.1. Socio-Demographic

Two-hundred and three (203) HEWs participated, with a response rate of 90.2% (203/225). Half of the HEWs (51.7%) were from the Gondar district. The average age (SD) and service year (SD) were 30.4 (±4.0) and 8.3 (±3.6) years, respectively. One-third (33%) had a college diploma. Nearly two-thirds (62.4%) were single. Forty (20.3%) had a family history of NCDs, with parents and siblings accounting for the vast majority (48.7% and 33.3%, respectively). Diabetes and hypertension were the most commonly reported NCDs in HEWs’ families (Table 1).

#### 3.1.2. Physical Activity

Of the total sample, 22.2% (n = 44) of HEWs reported engaging in at least 10 min of vigorous-intensity physical activity, and 92.9% (n = 170) engaged in 10 min of moderate-intensity physical activity in the past seven days. Only 63.7% (95% CI: 0.57, 0.71) of HEWs met the WHO physical activity recommendation of at least 150 min of moderate-intensity activity per week. The total reported time spent sitting per day was 93.67 min (Table 1).

#### 3.1.3. Dietary Habit

Self-reported fruit and/or vegetable (at least once a day) consumption was low (n = 40 (21.6%). Only 13% and 18.9% of HEWs consume fruits and vegetables at least once a day. A total of 135 (70.7%) HEWs reported using more than 5 g of salt daily. About one-fifth (n = 38) of HEWs consumed meals not cooked at home daily or weekly. Daily meat consumption was only reported in 11.7% (n = 40) of participants, whereas 27.6% consumed bread at least once daily (Table 1).

### 3.2. Level of NCD Knowledge

The mean (SD) comprehensive knowledge score of HEWs was 57.2 ± 5.39. Overall, 37.5% (95% CI: 30.8–44.2%) were in the highest tertile when combining general and specific NCD knowledge.

#### 3.2.1. General Knowledge of NCDs

On general NCD awareness questions, 36.9%, 32.6%, and 30.5% of HEWs had high, medium, and low levels of knowledge, respectively. The majority of participants classified cardiovascular disease, cancer, diabetes, and hypertension as NCDs. However, 47.3% and 54.7% of HEWs did not consider chronic respiratory disease and mental illnesses as NCDs.

A total of 43 (21.3%), 100 (49.5%), and 59 (29.2%) HEWs were classified as having a low, medium, and high level of knowledge of NCD risk factors. The vast majority (89.1%, 78.7%, 73.8%, and 100%, respectively) of HEWs were aware that unhealthy lifestyle behaviours, such as poor diets, physical inactivity, drinking and smoking, increase the risk of NCD. Two-thirds (66.8%) of HEWs did not identify stress as a risk factor, and one-third (34.2%) of HEWs did not recognise family history as an NCD risk factor.

A total of 169 (83.3%) HEWs believed carbohydrate-rich foods could be recommended for people with NCDs. One-fifth (20.2%) of participants reported that sugar-sweetened beverages are considered healthy options for people with NCDs (Table 2).

#### 3.2.2. Knowledge of Specific NCDs

Diabetes knowledge included a general awareness of the disease, its signs and symptoms, and complications. Knowledge of hypertension included general awareness of the condition, the diagnostic criteria, and its management. Cardiovascular disease (CVD) knowledge included a general awareness of CVD, its risk factors, and preventive strategies. Overall, 28.7%, 25.1%, and 46.2% of HEWs were categorized as having low, medium, and high levels of knowledge of diabetes, respectively. HEWs’ knowledge of hypertension was classified as low (23.9%), medium (54.7%), or high (21.4%). The study found that 15.1%, 49.0%, and 35.9% of HEWs had low, medium, and high knowledge of the cardiovascular disease, respectively. The detailed results for diabetes, hypertension and cardiovascular disease are provided in Appendix A.

As shown in Figure 1, one-third of HEWs lacked general awareness of NCD. Only half of HEWs had modest knowledge of NCD risk factors. The highest low level of knowledge was noted on diabetes mellitus.

### 3.3. Attitude towards Promoting Healthy Lifestyles

Overall, 52.2% (95% CI: 0.45, 0.59) of HEWs had a favourable attitude towards promoting healthy lifestyles in the community. Half of the HEWs strongly agree that advising clients to adopt healthier lifestyles is their routine task, while 18.2% disagreed. Over half of the HEWs stated they had little time to provide healthy lifestyle advice while doing routine tasks. Fifty-three HEWs believed their clients needed to be more open to receiving healthy lifestyle advice from HEWs. Forty HEWs reported that discussing healthy lifestyles was rewarding. Notably, 70 (34.5%) participants believed that the current healthcare practice limited their ability to engage in counselling (Table 3).

### 3.4. NCD Risk Perception

Half of the HEWs (51.2%) had a high perceived susceptibility to NCD. Most participants (81.8%) believed that changing one’s lifestyle could help prevent NCDs (Figure 2).

### 3.5. Perceived Self-Efficacy of NCD Health Promotion

Overall, more than half of HEWs (57.1%: 95% CI: 0.50, 0.64) reported feeling unconfident about delivering NCD health promotion. Over half of the HEWs felt unqualified to provide advice on physical activity and NCD preventive measures. Nearly half (46.8%) of HEWs were confident about measuring blood pressure, 65% of HEWs were unsure about measuring rapid blood glucose tests, and 62.1% were unsure about providing diabetic foot care advice (Figure 3).

### 3.6. Association of Self-Efficacy and NCD Risk Perception with Knowledge, Attitude, and Behavior

In the multivariable ordinal logistic regression analysis, perceived self-efficacy was significantly associated with higher NCD knowledge. HEWs with higher self-confidence in promoting healthy lifestyles had 2.21 times higher levels of NCD knowledge than HEWs with lower self-confidence (AOR: 2.21; 95% CI: 1.21, 4.07). Similarly, the higher perceived self-efficacy of HEWs was significantly associated with attitude (AOR: 6.27; 95% CI: 3.11, 12.61) and behaviour (AOR: 2.27; 95% CI: 1.08, 4.74).

Similarly, NCD risk perception of HEWs was significantly associated with the knowledge, attitude, and behaviour of HEWs. The knowledge of NCD among HEWs with high perceived NCD susceptibility was 1.89 times higher than that of HEWs with lower perceived susceptibility (AOR: 1.89; 95% CI: 1.04, 3.47). Perceived severity was also significantly associated with the level of NCD knowledge (AOR: 2.69; 95% CI: 1.46, 4.93). HEWs who perceived a high NCD susceptibility (AOR: 3.34; 95% CI: 1.63, 6.82) and benefits (AOR: 3.99; 95% CI: 1.64, 6.82) were more likely to engage in a sufficient level of physical activity (Table 4).

## 4. Discussion

We found insufficient NCDs knowledge, unfavourable attitudes towards promoting healthy lifestyles, insufficient physical activity, and poor dietary habits among HEWs. HEWs’ competencies were significantly associated with their perceived self-efficacy and NCD risk perception. The findings of this study are important for informing national and local health strategies related to NCDs health promotion in Ethiopia. This is because the World Health Organization (WHO) recommends task sharing and shifting NCD health services to community health workers to address the rising burden of NCDs and the shortage of skilled health workers. In addition, HEWs should have the necessary knowledge of NCDs, their risk factors, and the skills to promote healthy lifestyles for their integration into NCD services to succeed [7,8]. To the best of our knowledge, this is the first study that thoroughly evaluates the competency of Ethiopian HEWs’ knowledge of NCDs, attitudes towards their roles in advocating healthy lifestyles, personal lifestyle behaviours, and their association with self-efficacy and NCD risk perception. The study helps identify capacity building interventions that may be needed to support HEWS to work in their full potential.

With the increase in prevalence of NCDs in Ethiopia, HEWs should be well-trained to manage NCDs and their risk factors. Although most HEWs in this study recognized cardiovascular disease, cancer, diabetes, and hypertension as NCD categories, they failed to recognise chronic respiratory illness and cardiovascular disease as NCDs. Although chronic respiratory illnesses and cardiovascular disease are the two most frequent NCDs worldwide [1], a large majority of Ethiopian HEWs (46.3%, and 36%, respectively) did not consider them NCDs. Ethiopia has implemented a national action plan focusing on cardiovascular disease, diabetes, cancer, and chronic respiratory diseases [35]. The lack of knowledge of HEWs might impede the successful implementation of the action plan, as it showed that HEWs did not possess the necessary knowledge to be involved in NCDs health promotion.

Similarly, we found that two-thirds of HEWs did not identify adverse stress as a risk factor for NCD. Evidence indicates that stress-related metabolic syndrome often precedes NCDs, and stress-related NCDs are on the rise [36]. As a result, understanding stress as a risk factor for NCDs is vital for reducing NCDs in the community and among healthcare workers. A lack of NCD knowledge and its risk factors by HEWs has negative implications for their responsibilities in providing appropriate NCD information, advising on risk factors, and encouraging people to live a healthy lifestyle.

Our findings show that over half of the HEWs did not identify mental health as an NCD. This finding is concerning because mental illnesses, such as depression and anxiety, can significantly impact the community’s overall health and well-being. The ministry of health of Ethiopia has included mental, neurological, and substance use disorders in the national strategic plan for the prevention and control of major NCDs [35]. The UN high-level meeting on NCD in 2018 also advocates for mental health diseases to be classified as NCD. It also promotes mental ill health and NCD to be considered together, as they frequently share many characteristics [37]. The present finding was consistent with an Ethiopian study from five districts in the Jimma Zone, which found that 53.7% of HEWs described mental illness as a result of evil spirits’ possession [38]. A similar finding was seen among urban health extension workers in Addis Ababa, with 56% lacking adequate mental health illness knowledge [39]. Apart from the lack of mental health knowledge for HEWs, the norms, beliefs, and traditions that view mental illness as the result of an evil spirit could also be a reason. Due to these beliefs, many Ethiopians with mental illnesses rely on traditional healers and rituals rather than seeking the help of mental health professionals for their problems [38].

Overall, 29.9% (95% CI: 23.6–36.2%) of HEWs had a low level of NCD knowledge, 32.6% (95% CI: 26.2–39.0%) of HEWs had a moderate level of NCD knowledge, and 37.5% (95% CI: 30.8–44.2%) of HEWs had a high level of NCD knowledge. In contrast, a Nigerian study found that 23.0% of primary healthcare workers had adequate NCD knowledge [40], significantly lower than our findings. This variation could be due to knowledge score dimensions and classification differences. Despite differences in the percentage of NCD knowledge between studies, both studies revealed low NCD knowledge among healthcare workers, which is similar to previous findings from South Africa [41], Tanzania [42], and the Democratic Republic of the Congo (DRC) [43]. The Ethiopian strategic plan for preventing and controlling NCDs also stressed that low NCD awareness by healthcare providers remains the problem [35,44]. This common NCD knowledge could be remedied by providing more comprehensive training to HEWs, who often serve as the first point of contact for people seeking NCD support in the community. Moreover, this study found that HEWs lacked specific knowledge about diabetes mellitus and hypertension, which was consistent with a finding from South Africa [41,45]. HEWs’ lack of adequate knowledge of specific NCDs can significantly impact their roles in advising individuals for better health outcomes. This is because it is becoming more recommended that services for diabetes and hypertension be provided outside of the health facilities in communities [46,47].

Aside from the necessary knowledge of NCD, HEWs’ perception of their role in promoting healthy lifestyles is essential for effective NCD prevention and management. We found that 52.2% (95% CI: 0.45, 0.59) of HEWs had a positive attitude towards advocating health-promoting lifestyles to the community. A total of 47.8% of HEWs lacked the motivation to promote healthy living, which suggests that HEWs were not sufficiently active in NCD prevention and control. Although Ethiopia has expanded the scope of work of HEWs in the NCD program [44], a significant proportion of HEWs in this study did not consider healthy lifestyle promotion as their routine task. In this study, 18.7% of HEWs reported that not having enough time, the health system’s focus on communicable diseases, and the lack of people’s openness to accepting lifestyle advice all contributed to their poor attitude towards advocating for healthy lifestyles. Furthermore, more training, supervision, and support are needed to improve HEW motivation in health promotion to prevent and control NCDs.

Health professionals’ lifestyle behaviour is vital for effective NCD prevention and control, as it influences their counselling practice towards healthy lifestyles and motivates individuals to lead healthy lifestyles [15,16,17]. Moreover, it is expected that healthcare workers, including HEWs, would have healthier lifestyle habits compared to the general population. This expectation comes from the educational attainment and nature of their work, where healthcare professionals are exposed to health information and are likely to be more conscious of their health behaviour. Although healthcare workers are expected to be role models and are expected to practice what they preach to the community, three behavioural risk factors, namely physical inactivity, insufficient fruit and/or vegetable consumption, and excessive salt intake, were common among HEWs. Previous findings have shown that these unhealthy lifestyle choices among healthcare workers expose them to NCDs and impact their ability to promote healthy lifestyles to the public [17,48,49].

Our study found that 63.7% (95% CI: 0.57, 0.71) of HEWs had sufficient physical activity. Our HEWs were more physically active than Bangladeshi health workers, where 87% of health workers did not meet physical activity guidelines [50]. In contrast, the level of physical inactivity was higher than in the study from Sweden, where only 15% of healthcare workers reported a sedentary lifestyle [51]. Better-built environments in higher socioeconomic nations, such as sidewalks, bike paths, and parks, have encouraged physical activity by providing people with access to outdoor exercise spaces. This justification is also supported by previous studies in South Africa [52], China [53], and a systematic review in low- and middle-income countries [54].

Poor dietary habits have also been reported among HEWs in northwest Ethiopia. It could be attributed to limited access to nutritious foods and a need for more knowledge about proper nutrition. In this study, high salt intake and inadequate fruit and/or fruit consumption were identified among HEWs, which is consistent with a systematic review report [55], and previous studies in South Africa [56], Nigeria [57], Bangladesh [50], and India [58]. Our study found that 70.7% of HEWs consumed more than 5 g per day, which is above the WHO recommendation [59]. According to a systematic review of various nations, poor dietary habits, such as excessive salt intake and low fruit and vegetable consumption, were responsible for more deaths than smoking [55]. Evidence also showed that healthcare workers are becoming exposed to NCDs due to unhealthy lifestyle behaviours [60]. Moreover, health workers’ lifestyles not only expose them to NCDs but also impact their engagement in advocating for lifestyles to patients and the general public [48,49].

A present study found that higher NCD knowledge is associated with higher perceived self-efficacy and NCD risk perception. This association is likely because high self-confidence promotes better decision-making and better learning. Similarly, perceived susceptibility and severity showed a significant association with NCD knowledge. The perception of NCD vulnerability and severity could allow HEWs to learn more about the disease, its risk factors, and preventive strategies, allowing them to understand the disease better. Moreover, this study provides evidence of a strong association between a favourable attitude toward promoting healthy lifestyles and the HEWs’ confidence in their ability to promote them. Physical activity was significantly associated with self-efficacy, perceived NCD susceptibility, and perceived benefit of lifestyles. The higher odds of having a sufficient level of physical activity were noted among HEWs with a higher perceived self-efficacy (AOR = 2.27; 95% CI: 1.08–4.74), high NCD risk perception (AOR = 3.34; 95% CI: 1.63–6.82), and perceived benefits of a healthy lifestyle (AOR = 3.99; 95% CI: 1.64–6.82). This highlights the importance of improving self-efficacy and NCD risk perception behaviour of HEWs, as it is likely to motivate them to be physically active and adopt healthy lifestyles. The role of NCD training in improving knowledge, skill, self-efficacy, and changing behaviour was also highlighted in the Tanzania pilot training for public health workers [61]. However, the authors thought that improving self-efficacy and NCD risk perception can motivate HEWs to engage in physical activity; the relationship in the opposite direction might be possible, which needs further empirical evidence.

The findings of this study have implications for future research as well as community-based NCD health promotion practices, policy, and program. First, community health workers play an important role in health education and in promoting healthy lifestyles. Their self-efficacy in NCD health promotion and risk perception, on the other hand, can all impact on the effectiveness of their role. This suggested that health professionals’ NCD risk perception to influence health behaviour is more effective when their self-efficacy in their ability to promote health promotion and lifestyle behaviour is sufficient. In addition, community health workers’ personal lifestyle choices are critical in leading the community by example for improving the health outcomes of individuals and communities. Our study highlights the importance of providing education and training on NCD prevention and management, and prioritizing healthcare professionals’ personal health and well-being. The findings of this study can inform the development of targeted interventions to improve the self-efficacy, NCD risk perception, and lifestyle behaviour of community health workers, which in turn can lead to better NCD health outcomes. Future research (both qualitative and quantitative) on the challenges of engaging community health workers in NCD health promotion and training needs, as well as the impact on their capacity in NCD prevention and control, is also strongly recommended.

## 5. Limitations

Even though the study attempted to show the competency of HEWs in NCD knowledge, attitude, and personal lifestyle behaviour, it was not free from limitations. Since the study was cross-sectional, a causal relationship between variables might not be possible. Moreover, the self-reported nature of responses might introduce self-reported and social desirability bias. Furthermore, no comparisons were made regarding knowledge, attitude, and personal lifestyle behaviour between HEWs and other health professionals or the general public.

## 6. Conclusions

Despite HEWs being an important component of community-based NCD health promotion, their NCD knowledge, attitude, and lifestyle behaviour were lower than expected.

We found that NCD knowledge and attitude towards health promotion were significantly associated with HEWs’ perceived self-efficacy and NCD risk perception.

Moreover, insufficient physical activity and unhealthy dietary habits, such as insufficient fruit and/or vegetable consumption, was a problem among HEWs, who are assumed to be role models for leading the community by example. HEWs’ levels of physical activity were significantly associated with their perceived self-efficacy, NCD risk perception, and perceived benefits of lifestyle change.

Our participants’ NCD knowledge, attitude, and personal lifestyle behaviour suggest that community health workers (HEWs) would benefit from additional capacity building, which includes interventions to improve their lifestyle behaviours, NCD risk perception, and self-efficacy for NCD health promotion.

While HEWs have successfully improved maternal, child, and infectious disease outcomes in Ethiopia over the last few decades, their role should be reconsidered in light of the new NCD health demands. HEWs have the potential to participate in the promotion of healthy lifestyles, provided that they are adequately trained and supervised by the health system.

## Figures and Tables

**Figure 1 ijerph-20-05642-f001:**
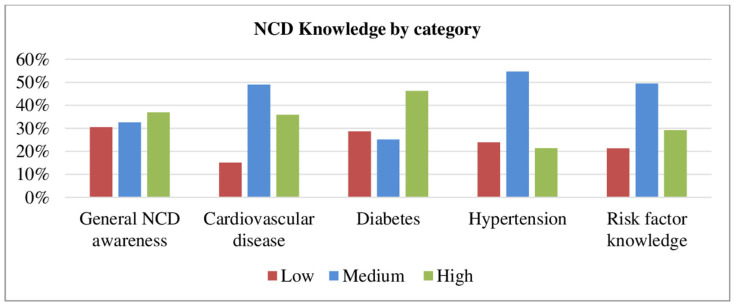
NCD knowledge by category of HEWs in northwest Ethiopia, 2022.

**Figure 2 ijerph-20-05642-f002:**
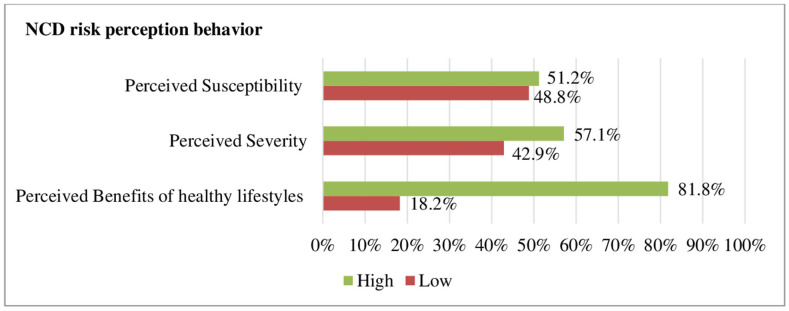
NCD risk perception behaviour of HEWs in northwest Ethiopia, 2022.

**Figure 3 ijerph-20-05642-f003:**
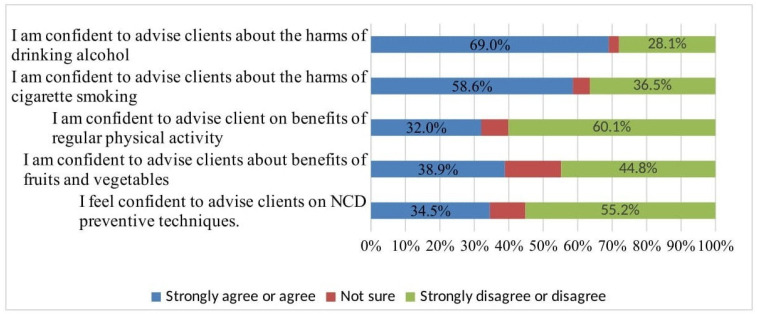
Self-efficacy of HEWs on promoting health lifestyles in northwest Ethiopia, 2022.

**Table 1 ijerph-20-05642-t001:** Socio-demographic and lifestyle behaviours of Heath Extension Workers (HEWs) in Northwest Ethiopia, 2022.

Variables		Frequency (N)	Percentage (%)
Mean age ± standard deviation (SD)	30.4 ± 4 years		
Mean service year ± SD	8.3 ± 3.6 years		
Residence			
Gondar		105	51.7
Dabat		55	27.1
Debark		43	21.2
Educational status			
Certificate Level 3		16	7.9
Certificate Level 4		63	31.0
Diploma		67	33.0
First degree		57	28.1
Marital status			
Living with a partner		134	66.3
Living on their own		68	33.7
Family history with NCDs			
Yes		40	20.3
No		102	51.8
Not sure		55	27.9
Physical activity measure		Mean	(SD)
Walking MET—min/wk		144.5	49.7
Moderate MET—min/wk		617.2	242.5
Vigorous MET—min/wk		400	198.0
Sitting time per day (in minute)		93.67	39.59
Sitting time per week (in minute)		171.08	107.03
		Frequency	Percentage
Sufficiently physically active		121	63.7
Insufficiently physically active		69	36.3
Dietary habits			
Fruit and/or vegetables (serves per day) (n = 185)		40	21.6
Salt consumption per day (n = 195)			
Less than 1 tea spoon (<5 g)		56	29.3
Greater than 1 tea spoon (>5 g)		135	70.7
Meals not prepared at home (n = 196)			
Never		66	33.7
Daily or weekly		38	19.4
Monthly		92	46.9

Note: IPAQ, International Physical Activity Questionnaire; MET—min/wk, metabolic equivalent minutes per week; NCD, non-communicable disease; SD, standard deviation.

**Table 2 ijerph-20-05642-t002:** General knowledge on non-communicable diseases (NCD) and risk factors among HEWs in northwest Ethiopia, 2022.

Variable	Responses	
	Correct (%)	Incorrect (%)
Types of NCDs		
Cardiovascular disease (CVD)	130 (64.0)	73 (36.0)
Cancer	148 (72.9)	55 (27.1)
Chronic respiratory disease	109 (53.7)	94 (46.3)
Diabetes mellitus	203 (100)	0 (0)
High blood pressure	203 (100)	0 (0)
Mental illness	92 (45.3)	111 (54.7)
Risk factors of NCD		
Age	168 (83.2)	34 (16.8)
Family history of NCD	133 (65.8)	69 (34.2)
Low fruits and vegetables consumption	180 (89.1)	22 (10.9)
High sugar intake	148 (73.3)	54 (26.7)
High salt intake	171 (84.7)	31 (15.3)
Being overweight or obesity	181 (89.6)	21 (10.4)
Physical inactivity	159 (78.7)	43 (21.3)
Smoking	202 (100)	0 (0)
Alcohol	149 (73.8)	53 (26.2)
Stress	67 (33.2)	135 (66.8)
NCD prevention		
Avoiding smoking	180 (88.7)	23 (11.3)
Regular exercise	181 (89.2)	22(10.8)
Healthy diet	185 (91.1)	18 (8.9)
Limiting alcohol consumption	185 (91.1)	18 (8.9)
Dietary recommendations		
High fat foods	19 (9.4)	184 (90.6)
Soft and energy drinks	41 (20.2)	162 (79.8)
High fibre foods	166 (81.8)	37 (18.2)
Carbohydrate-rich foods	169 (83.3)	34 (16.7)
Fruits and vegetables	189(93.1)	14 (6.9)

**Table 3 ijerph-20-05642-t003:** Attitudes of HEWs towards promoting healthy lifestyle measures in northwest Ethiopia, 2022.

Attitude of Promoting Lifestyles		Frequency		Percentage	
Favourable attitude		106		52.2	
Unfavourable attitude		97		47.8	
Questions (N = 203)	Strongly agree (%)	Agree (%)	Not sure (%)	Disagree (%)	Strongly disagree (%)
Advising clients to adapt healthier lifestyles is part of my routine task	104 (51.2)	44 (21.7)	18 (8.9)	22 (10.8)	15 (7.4)
Discussing lifestyle is useful to improve people’s health	138 (68)	65 (32)	0 (0)	0 (0)	0 (0)
Discussing lifestyle is more useful to people with NCDs	111 (54.7)	45 (22.2)	6 (3.0)	27 (13.3)	14 (6.9)
I believe my clients expect me to discuss about lifestyle	92 (45.3)	60 (29.6)	35 (17.2)	7 (3.4)	9 (4.4)
I do not have adequate time to provide counseling on healthy lifestyle during routine home-to-home visit	53 (26.1)	56 (27.6)	28 (13.8)	43 (21.2)	23 (11.3)
Clients are not receptive to receiving healthy lifestyle counseling	28 (13.8)	25 (12.3)	17 (8.4)	100 (49.3)	33 (16.3)
Discussing healthy lifestyle behaviors with clients is rewarding	67 (33.0)	53 (26.1)	43 (21.2)	22 (10.8)	18 (8.9)
The current health care structure limits my ability to provide lifestyle advice	32 (15.8)	38 (18.7)	30 (14.8)	80 (39.4)	23 (11.3)
Knowing more about healthy living will help me counsel clients.	119 (58.6)	67 (33.0)	17 (8.4)	0 (0)	0 (0)

**Table 4 ijerph-20-05642-t004:** Association of self-efficacy and NCD risk perception with knowledge, attitude, and behaviour of HEWs in northwest Ethiopia, 2022.

Variables		NCD Knowledge		Attitude		Physical Activity	
		AOR (95% CI)	*p*-Value	AOR (95% CI)	*p*-Value	AOR (95% CI)	*p*-Value
Perceived Self-efficacy							
Higher confident		2.21 (1.21, 4.07)	0.010	6.27 (3.11, 12.61)	<0.000	2.27 (1.08, 4.74)	0.030
Lower confident		1		1		1	
NCD risk perception							
Perceived susceptibility	High	1.89 (1.04, 3.47)	0.038	0.49 (0.24, 0.98)	0.042	3.34 (1.63, 6.82)	0.001
	Low	1		1		1	
Perceived severity	High	2.69 (1.46, 4.93)	0.001	1.79 (0.93, 3.43)	0.079	1.04 (0.51, 2.10)	0.923
	Low	1		1		1	
Perceived Benefits	High	1.92 (0.89, 4.15)	0.095	1.27 (0.56, 2.84)	0.565	3.99 (1.64, 6.82)	0.001
	Low	1		1		1	

Note: The models were adjusted for age, highest level of education, and service years.

## Data Availability

The dataset used for this research will be available upon request of the principal investigator.

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
