# Peer review of "An Evaluation of Community Health Workers’ Knowledge, Attitude and Personal Lifestyle Behaviour in Non-Communicable Disease Health Promotion and Their Association with Self-Efficacy and NCD-Risk Perception"

_ijerph, 2023, doi:10.3390/ijerph20095642_

Round 1

Reviewer 1 Report

Dear Authors,

The subject of the article is very interesting, and it can benefit health promotion. My congratulations for the work done.

However, from my perspective, it would be necessary to make some adjustments. Below you can read my comments after carefully reviewing your paper:

  • The manuscript has numerous typos and punctuation errors. For instance: line 60, 129, 135, 159, etc. It needs to be fully reviewed.

  • Firstly, the abstract needs to be improved, because it does not completely reflect the content of the entire manuscript. It contains many results, but few implications or more in depth statements.

  • Line 35: behavior is not an adequate keyword for this paper. I would change it for lifestyle. 

  • Lines 68-71: “Furthermore, community health care workers are seen as a role model by the community and are expected to lead by an example. Hence, it is important that these community health care workers engage in healthy lifestyle behaviors such as physical activity and a healthy diet and improve their personal lifestyle behavior [20-22].” The bibliography provided refers to physicians and how their orientations are influenced by their own health practices. But does the same happen with community health care workers? What is the inference based on when equating physicians with health extension workers? From my perspective, your should justify this inference. 

  • Line 96: You write about the “recruitment” of rural HEWs, but you do not give information about the recruitment; you describe the inclusion or exclusion criteria of the participants. The same is described for urban HEW. Recruitment information is pending.

  • Lines 11-122: Authors should provide some more information about the questionnaire, especially those that have to do with the psychometric characteristics. For instance, could you specify its content validity, or the reliability and reproducibility of the instrument? Moreover, why have you collected questions from different sources and generated a new questionnaire? Is it so? Could you specify which part of the questionnaire is due to each source?

  • Line 129: Specify what MET is (Metabolic Equivalent of Task).

  • Line 303: The year is given, when it is not provided in later tables and figures. Homogenize them.

  • Line 308: The type of font for the description of the figure is different from that of figure 1. Review formats of the entire manuscript.

  • Line 317-318: the heading is in bold. It would be good to modify the format of the titles, because it is confusing. There are several formats, and you don't know which one is the main one and which are the secondary ones.

  • Lines 370-371: Mental health is not identified as a NCD by WHO. What is the basis for doing so and including it as an NCD?

  • Line 416: How does reference 54 differ from references 20-22 that have been provided in the Introduction? Why are these not taken into account now? If, as stated in the Introduction and now recalled here, “Health professionals’ own personal lifestyle behavior is vital for effective NCD prevention and control, as it influences their counseling practice towards healthy life styles and motivates individuals to lead healthy lifestyles [54].” Why do include authors a third reference (54), and they don’t discuss references 20-22?

  • Lines 462-464: the conclusion you draw here can be discused. They comment that if we improve self-efficacy and NCD risk perception, then HEW will feel more motivated to engage in physical activity and to lead a healthy lifestyle. But couldn't it be the other way around? Why does the relationship have that direction, and not the other way around? Could it not be that those who perform physical activity and have a healthy lifestyle have a higher perception of self-efficacy and risk of NCD?

  • Lines 464-466: The Discussion section ends with a statement about Tanzania and public health workers. It would be more appropriate to close this important section in another way, with a more deep statement, or to summarize what has been contributed.

  • Lines 482-483: The research is based on a fundamental assumption, and it is now also collected here: the HEW are an example, or model, that lead the community with example. But, for instance, in line 490 you write that HEWs have done very well years before with maternal/child diseases. Does that mean that the HEWs were women who had also been mothers? Is it a necessary condition to carry out your function as community health workers? According to the authors, I understand that the answer would be affirmative. Is it so?

  • Nothing is said about the passing of the questionnaire. How was it made? When? How many participants were invited and how many responded? How long did it take? Was it done individually or in a group? These and other issues should be specified in the paper.

  • From all the data you have, which is a lot, you draw few conclusions. In fact, you establish very few relationships between all the variables you have handled. Why? Without going beyond a descriptive analysis, you could have found suggestive data linking, for example, personal lifestyle behavior and knowledge in HEW, or between attitude and personal lifestyle, etc.

  • The conceptual implications could be elaborated in more depth. Moreover, it would be great to add practical implications that could be illustrated with detailed examples from health practice.

  • In the same way, in the Discussion section, only conclusions are given as were given in the Results section, but their implications are not developed. This section should discuss the implications of the findings in context of existing research, but the link between them is limited to comparisons between countries. You should choose the results that are important and “discuss” their implications. In this section we should explain what is new in the study, or why this study is important.

  • The References section has double numbering.

Reviewer 2 Report

YOU MIGHT CONSIDER REMOVING SOME OF THE BIBLIOGRAPHIC REFERENCES, THEY ARE TOO MANY AND LOSE IMPORTANCE THE OTHERS

Author Response

Thank you for your constructive feedback.

We removed some bibliographic references. 

We revised typos and punctuation errors throughout the manuscript. The paper is also reviewed for writing style and spelling.  (please see the track change document attached)

Reviewer 3 Report

I consider the problem of the article very appropriate, health promotion is fundamental, as it brings many health gains for the population in general and for the management bodies. Identifying the factors that influence the promotion of behaviors and healthy lifestyles among community health agents is paramount, as they must be an example in the community where they work. The manuscript is presented in accordance with the guidelines of the International Journal of Environmental Research and Public Health (IJERPH). It has some methodological weaknesses, I suggest an in-depth review of the methodology. 

Round 2

Reviewer 3 Report

The authors took into account all suggestions for improvement from the first review, incorporating the changes in this version of the article. I consider that the article is suitable for publication.